# NMR Analysis Suggests Synergy between the RRM2 and the Carboxy-Terminal Segment of Human La Protein in the Recognition and Interaction with HCV IRES

**DOI:** 10.3390/ijms24032572

**Published:** 2023-01-29

**Authors:** Aikaterini I. Argyriou, Georgios A. Machaliotis, Garyfallia I. Makrynitsa, Eleni G. Kaliatsi, Constantinos Stathopoulos, Georgios A. Spyroulias

**Affiliations:** 1Department of Pharmacy, University of Patras, GR-26504 Patras, Greece; 2Department of Biochemistry, School of Medicine, University of Patras, GR-26504 Patras, Greece

**Keywords:** La, lupus antigen, RNA-binding proteins, hepatitis C virus, IRES, RNA virus, NMR

## Abstract

The La protein (lupus antigen) is a ubiquitous RNA-binding protein found in all human cells. It is mainly localized in the nucleus, associates with all RNA polymerase III (Pol III) transcripts, as the first factor they interact with, and modulates subsequent processing events. Export of La to the cytoplasm has been reported to stimulate the decoding of specific cellular and viral mRNAs through IRES-dependent (Internal ribosome entry site) binding and translation. Using NMR (Nuclear Magnetic Resonance) spectroscopy, we provide atomic-level-resolution structural insights on the dynamical properties of human La (hLa) protein in solution. Moreover, using a combination of NMR spectroscopy and isothermal titration calorimetry (ITC), we provide evidence about the role and ligand specificity of the C-terminal domain of the La protein (RRM2 and C-terminal region) that could mediate the recognition of HCV-IRES.

## 1. Introduction

The La protein (lupus antigen) is an important RNA-binding protein (RBP) and belongs to the La-related proteins (LaRPs) family [1]. It was first described as a cytoplasmic RNA protein antigen in the sera of patients with systemic lupus erythematosus (SLE) and Sjögren’s syndrome and has been studied extensively, because it is the first protein that all RNA Polymerase III (Pol III) transcripts encounter upon their synthesis [2,3,4]. The La protein binds the 3′ end of all pre-mature Pol III transcripts to assist proper folding and subsequent maturation. Due to its role in transcription regulation, the La protein is mainly localized in the nucleus, though in specific cases it can shuttle to the cytoplasm and interact with a variety of different mRNAs (endogenous and viral) to stimulate their translation [5,6,7]. For example, it was reported recently that La overexpression can stimulate cap-independent translation through direct binding on the IRES of reporter genes [7]. Interestingly, it was also reported that the La protein acts as the gatekeeper for specific pre-tRNAs, to prevent their processing by Dicer for the production of specific miRNAs [8]. These observations highlight the alternative roles of the La protein and its contribution to important RNA–protein interactions.

The human La protein is a multi-domain protein, consisting of the La motif (LaM) and two RNA recognition motifs (RRM1 and RRM2). LaM and RRM1 are connected with a small linker (with different sizes among species) and constitute the N-terminal domain (NTD), while RRM2 is located at the C-terminal domain (CTD) of the protein [9]. Previous studies have shown that LaM and RRM1 (known as the “La Module”) cooperate to bind to the UUU-3′-OH sequence at the 3′ end of the nascent transcripts of RNA Polymerase III to ensure protection from exonucleases, ensuring accurate subsequent maturation and folding [10,11]. The C-terminal domain of the La protein is the least-conserved region of all the LaRP proteins, with an elusive role so far. It consists of an atypical RRM domain (RRM2), a short basic motif (SBM), a nuclear localization signal (NLS) and a nuclear retention element (NLE) [12,13,14]. It has been proposed that the C-terminal domain of the La protein mediates the interactions that stimulate the internal ribosome entry site (IRES)—the mediated translation of some endogenous mRNAs and mRNAs that contain the HIV TAR element as well as viral RNAs (e.g., the hepatitis C virus and poliovirus) [15,16,17,18,19,20,21]. The mechanism by which the La protein recognizes the IRES sequence and promotes translation is not well understood. Isothermal titration calorimetry (ITC) experiments using the IRES sequence of the hepatitis C virus (HCV) genome and different mutant domains of the human La have revealed that the cooperation of LaM-RRM1 is not sufficient for the recognition of this sequence. Nevertheless, it is not clear whether the recognition of IRES is only due to the C-terminal domain of the La protein or the cooperation of both the N-terminal and the C-terminal domains. Experiments suggested that the La protein uses a different mode for the recognition of IRES, compared to the recognition of UUU-3′-OH in the nascent transcripts where LaM and RRM1 suffice [6,22].

Given the important cellular role of the La protein and the fact that absence of the La protein causes disruption of the nucleolar organization, the recognition of various RNA structured domains by the La protein as a whole or through synergy between its different modules still awaits clarification [7]. The solution structures of each domain (LaM PDB ID:1S7A, RRM1 PDB ID: 1S79 and RRM2 PDB ID: 1OWX) have been individually determined through NMR. The crystal structure of the “La Module” in complex with RNA oligos was also determined and reported in the literature (PDB ID: 2VOD, 2VOO, 2VOP, 2VON, 1ZH5 and 1YTY) [9,10,11,22,23]. In addition, we have determined that the La motif alone is capable of binding precursor tRNAs, which are among the in vivo ligands of the La protein [7]. Interestingly, the recent demonstration of an La protein representative in a *Tetrahymena* that lacks any recognizable RRM motif confirms previous observations and underlines the necessity of studying the individual domains and modules of the La protein in their biological context [24].

The current knowledge suggests that the presence of all three structured domains of the La protein are necessary for HCV-IRES binding [6,7]. To elucidate the role of the RRM2 domain and the unstructured C-terminal region of the human La (the region spanning amino acids 225–408) in HCV-IRES recognition, we combined NMR analysis with the thermodynamics measurements of the possible binding affinities. Toward this direction, we investigated the interaction of the IV domain of HCV-IRES with La 224–359 (RRM2-SBM), La 224–408 (RRM2-Cter), La 105–359 (RRM1-RRM2-SBM) and La 105–408 (RRM1-RRM2-Cter). Finally, the interaction of La 224–334 (RRM2) with a polyU_10_ ligand was also studied via NMR. Taken together, our results suggest that the RRM2 domain exhibits specificity for HCV-IRES RNA recognition, a new and important role for this overlooked domain of the La protein.

## 2. Results

### 2.1. NMR Conformational Study of Human RRM2-Cter (La 225–408)

Although the resonance assignment for the La RRM2 is reported in the literature (BMRB accession no 5235), the NMR resonance assignment of a larger polypeptide, which spans the RRM2 and the C-terminal (Cter) segment, is not available. The ^1^H-^15^N HSQC spectrum of RRM2-Cter (La 224–408) (Figure 1A) indicates a well-folded protein with a stable tertiary structure in solution. An analysis of the 3D heteronuclear NMR spectra resulted in the assignment of 90% of the backbone nuclei and 62% of the side chain nuclei (Figure 1). All ^1^H, ^15^N and ^13^C chemical shifts of the RRM2-Cter were deposited in the BioMagResBank (http://www.bmrb.wiesc.edu; accessed on 29 November 2022) under accession no. 51713. No backbone amide resonances could be detected for residues M224, S225, L226, F236, E300, I319, R334, R335, F336, K339, G340, K341, G342, N343, K344, F357, A378, D407 and Q408. Most of the unassigned residues are in the C-terminal region of the protein, while M224, S225, L226, F236, E300, I319 and R334 are the unassigned residues of the RRM2 domain (Figure 1B). The fact that most of the unassigned residues are in the structureless and flexible C-terminal region leads to signal overlap in the center of the ^1^H-^15^N HSQC spectra (Figure 1A).

The secondary structure of RRM2-Cter, produced by the TALOS+ server (Appendix A), shows that the RRM2 domain is structured, while the C-terminal region is unstructured. The topology of this polypeptide predicted by TALOS+ is *β-α-β-β-α-β-β-α*. Based on the chemical shifts difference of the C_β_ and C_γ_ of the prolines, all prolines (P348, P381 and P394) are in trans conformation [25].

The model of RRM2 linked to the C-terminal polypeptide (La 335–408; 74 residues) is determined using CS Rosetta (PDB: 1OWX is the available NMR model only for the RRM2 domain). This RRM2-Cter model is used in the present study to map the La’s residues involved in the interaction with IRES (Figure 1B and Appendix A) [26,27,28,29].

### 2.2. ^15^N Relaxation Study of RRM2-Cter (La 225–408)

The average values of *R*_1_, *R*_2_ and *heteroNOE* (Table 1), derived by the analysis of the ^15^N relaxation data for RRM2-Cter as well as the values of *R*_2_*/R*_1_ and *R*_1 ×_
*R*_2_ (Figure 2 and Appendix A), demonstrates the rigid core of the RRM2 domain and the highly flexible C-terminal region in the ps–ns time scale.

Going through the analysis of the results of *R*_2_, it seems that some residues exhibit high *R*_2_ values (Appendix A). This may suggest that these residues are involved in a conformational exchange. According to the *R*_2_*/R*_1_ and *R*_1_
*× R*_2_ analysis (Figure 2), the same residues deviate from the average values, further supporting a possible exchange equilibrium between the different conformers. Specifically, the *R*_1_
*× R*_2_ values for the residues L240, D242, R246, N255, K269 and E270 range from ~24 to 36 (average value ~18). These residues are located either in the *β*1-*α*1 and *β*2-*β*3 loops (L240, D242, K269 and E270), or in *α1* close to these loops [30], which are linkers of the elements (*β*2, *β*3 and *α*1) that comprise the RNA binding surface. These data suggest that these RRM2 segments possess a conformational plasticity.

### 2.3. NMR Titration Experiments Using Human La Domains and HCV-IRES IV Domain

The interaction of the RRM2-Cter of the La protein with the IV domain of IRES of the hepatitis C virus (HCV-IRES) was monitored through NMR titration experiments. To identify the specific interaction sites of RRM2-SBM (La 224–359), RRM2-Cter (La 224–408), RRM1-RRM2-SBM (La 105–359) and RRM1-RRM2-Cter (La 105–408), increasing amounts of unlabeled HCV-IRES were added to the ^15^N-labeled La polypeptides. The ^1^H-^15^N HSQC spectra were recorded after each addition, and the chemical shift changes (chemical shift perturbation; CSP) were extracted. During titration, some of the NH signals are shifted, while others disappear probably due to exchange broadening, suggesting an interaction between the human La polypeptide and the IRES domain of HCV. NH resonance shift and disappearance are typical features of a chemical exchange process among two interacting biomolecules, which undergo a fast-to-intermediate exchange between the free and bound form of the polypeptide (Appendix A).

#### 2.3.1. Interaction of RRM2-SBM (La 224–359) and RRM2-Cter (La 224–408) with IRES

NMR titration experiments were applied to investigate the binding properties of RRM2-SBM (La 224–359) and RRM2-Cter (La 224–408) toward the IRES domain IV of HCV. The ^1^H-^15^N HSQC spectra of the RRM2-SBM and RRM2-Cter after the addition of the HCV-IRES IV domain in 1:1 ratio are shown in Appendix A, respectively. For both experiments, the CSP values were calculated using Equation (1) (Section 4.3). For RRM2-SBM:HCV-IRES, the NMR analysis revealed that 25 NH signals are affected after the addition of HCV-IRES in the protein sample. From these, 19 exhibited CSP values greater than 0.076 ppm, which is the threshold, determined by 2× the standard deviation value (σ) calculated for all CSPs. The NH peaks of six residues disappeared during the titration experiment, possibly due to the exchange broadening (Figure 3B and Appendix A). The NMR analysis of RRM2-Cter after the addition of the HCV IRES IV domain revealed that 31 NH signals are affected, with 19 of them exhibiting CSPs greater than the threshold (0.064), and the NH peaks of 12 residues disappeared during the titration experiment (Figure 4B and Appendix A).

Mapping the perturbated and disappeared NH of the La amino acids on the surface of the 3D model (Figure 3A and Figure 4A) suggests that most of them belong to the *β*2 and *β*3 strands and the *α*1 and *α*3 helixes for both the La polypeptides. These secondary structure elements are in close proximity and define the interaction area of the RRM2 for HCV-IRES. Even if the human La RRM2 exhibits the structural characteristics of the typical RRM domain architecture, its interaction mode with HCV-IRES seems to vary from previously reported studies, suggesting a new interaction mode with IRES [6]. Specifically, the interaction surface of the RRM2-SBM/RRM2-Cter does not involve the core of the RRM2 domain (*β*1 and *β*3 strands), like other typical RRM domains, rather it involves the *β*2 and *β*3 strands, which lie on one site of the domain (Figure 3A and Figure 4A). When the entire C-terminal polypeptide is used for the interaction, 6 more amino acids seem to disappear (12 in total) during the titration studies, compared with the shorter polypeptide (spanning only the RRM2 domain). The interaction site of RRM2-Cter contains more amino acids from the *α*1-helix, *β*1*α*1-loop, *β*2*β*3-loop, *β*3-strand and the first turn of the *α*2-helix (Figure 4A).

The ^1^H-^15^N HSQC peaks of Trp261 (*β*2-strand), Phe264 (*β*2*β*3-loop) and His250 (*α*1-helix), that is, the aromatic amino acids located at the RNA interaction site, are affected or disappear upon HCV-IRES addition. Phe275 (*β*3-strand) is affected only when the whole C-terminus polypeptide is present. Not only the aromatic amino acids but also other amino acids of different types are affected or disappeared upon the addition of HCV-IRES. The aromatic residues, along with the arginines and the glutamic and aspartic acids can establish π–cation interactions with the bases of the RNA [31]. Other amino acids such as leucines and isoleucines can form hydrogen bonds with the RNA among KH domains. Van der Waals interactions, which are weaker, can also occur between RRM2 and HCV-IRES. According to the NMR data, more residues seem to become involved in the interaction of that La RRM2 that is bearing the entire C-terminal sequence, which suggests that the unstructured C-terminal has a role in the recognition and selectivity of HCV RNAs.

#### 2.3.2. Interaction of RRM1-RRM2-SBM (La 105–359) and RRM1-RRM2-Cter (La 105–408) with IRES

An important factor in the function of the La protein is whether the two RNA recognition motifs, RRM1 and RRM2, act in synergy to recognize, select and bind specific RNA. To elucidate the potential cooperation of RRM2 with RRM1 (a part of the N-terminal domain of the La protein) in the La protein binding to HCV-IRES, we attempted to study the interaction of RRM1-RRM2-SBM (La 105–359) and RRM1-RRM2-Cter (La 105–408) with the IRES of HCV (Appendix A, respectively).

The NMR analysis of the titration between RRM1-RRM2-SBM and HCV-IRES in a 1:1 ratio (Figure 5B) revealed that the CSP values (Equation (1)) of 26 NHs were above the threshold value, which is 0.04 (calculated as described above), and 43 NH peaks disappeared due to the addition of HCV-IRES, with most of the affected NHs belonging to RRM2 residues. Specifically, 20 out of 69 affected residues are located in the RRM1 domain, 7 in the linker between RRM1 and RRM2, and 42 in the RRM2-SBM region (Figure 5A and Appendix A).

These NHs affected by HCV-IRES are mapped on the surface residues of the individual models of RRM1 (PDB ID: 1S79) and RRM2-Cter, as calculated by CS Rosetta. To investigate whether the linker between RRM1 and te RRM2 is involved in the interaction of HCV-IRES, these residues are also mapped to the AlphaFold 3D model of the full-length La protein [32,33] (Appendix A). The RRM1 residues, which disappear or exhibit the largest CSP values, are associated with the *β*-strands (*β*1, *β*2, *β*3 and *β*4) and the *α*2 and *α*3 helixes. Specifically, these residues are in the regions of the *β*-strands close to the *a*2 and *a*3 helixes, while the rest of the molecule, the *α*1 helix and the *β*1*α*1, *β*2*β*3 and the *α*2*β*4 loops remain unaffected (Figure 5A(i)). Noteably, RRM1 seems to utilize a surface that spans the entire *β*-sheet in the HCV-IRES RNA interaction, instead of the *β*1- and *β*3-strands that comprise the typical surface usually involved in RNA binding.

Mapping of the perturbated and disappeared NH resonances of the RRM2-SMB, as a result of the interaction of RRM1-RRM2-SBM with HCV-IRES, indicates that most of the residues are on the *beta* strands surface of RRM2 and in the *α*3-helix of RRM2 (Figure 5A(ii)). Specifically, 11 out of the 42 affected residues of RRM2-SBM are located in the *β*-sheet, and 10 out of the 42 affected residues are found in the *α*3-helix. The interaction site of RRM2 remains the same compared to that identified during the RRM2-SBM interaction with HCV-IRES. According to our NMR data for the RRM1-RRM2-SBM polypeptide, the RNA interaction is stronger compared to the NMR data obtained during the titration of RRM2-SBM with HCV-IRES, underlying the impact of RRM1 in the above process. Indeed, more NH peaks disappear at the ^15^N-HSQC, upon the HCV-IRES addition, compared to the titration of RRM2-SBM with HCV-IRES.

We also attempted to study the interaction between the RRM1-RRM2-Cter polypeptide and HCV-IRES by NMR. However, the vast majority of the amino acids’ NHs disappear upon the addition of HCV-IRES, preventing the observation and evaluation of the chemical shift changes. Additionally, the interaction of the two biomolecules results in the formation of insoluble aggregates, which reduces the concentration of the biomolecular complex in the NMR sample, further preventing the observation of the low-intensity NHs resonances and their chemical shift changes (Appendix A). Thus, we assume that the residues of the whole polypeptide are directly affected by the RNA interaction.

### 2.4. NMR Titration Experiments of Human La RRM2 with PolyU_10_

The role of the La protein in binding the UUU-3′-OH termini of the RNA polymerase III nascent transcripts during their maturation proccess is well-established, but the role of the RRM2 to this process is still debated. To investigate whether RRM2 has an impact on this process or exhibits any specificity on IRES recognition, we performed an NMR titration experiment to study the possible interaction of RRM2 (La 224–334) with the polyU_10_ sequence.

The overlay of the ^1^H-^15^N HSQC of RRM2 (blue) and the RRM2:polyU_10_ complex in a 1:1 ratio (Appendix A) shows that only minor chemical shift changes are observed after the addition of the RNA sequence. The CSP analysis revealed that all the CSP values are low, while only seven residues have CSP values (Equation (1)) above the threshold (Figure 6B), ranging from 0.019 ppm to 0.058 ppm. These amino acids are found in the *β*2-strand, the *β*2*β*3-loop and in the first turn of *α*1-helix, suggesting that RRM2, as a stand-alone polypeptide, does not show any significant interaction with polyU_10_, which is the physiological ligand of the La protein (Figure 6A). The intensity changes of the peaks during the titration were also examined, but no meaningful changes were observed (Appendix A) in most of the residues, with the exception of the few regions where CSPs values above average were observed. In total, all these data provide experimental evidence for a weak or minor interaction between RRM2 and the polyU10 oligomer (Appendix A).

### 2.5. Thermodynamics of the Interaction between La Domains and IRES

The interactions between the different La domains (RRM2-SBM, RRM2-Cter, RRM1-RRM2-SBM and RRM1-RRM2-Cter) and IRES-HCV domain IV were studied by using isothermal titration calorimetry (ITC) to determine the dissociation constants. Two sets of experiments were performed for each interaction study, titrating the La polypeptides with IRES-HCV domain IV. All the polypeptides containing the RRM2 domain do interact with HCV-IRES RNA (Figure 7). Out of all the La’s polypeptides containing the RRM2 domain, RRM2-SBM displays the highest K*_D_* (K*_D_* = 28.2 ± 1.21 μΜ, Table 2).

The rest of the three polypeptides, which contain the entire C-terminus region and/or the RRM1, exhibit higher affinity for IRES-HCV. Indeed, the K*_D_* for RRM2-Cter is 15.3 ± 1.49 μΜ, for RRM1-RRM2-SBM is 10.3 ± 2.04 μΜ and for RRM1-RRM2-Cter is 9.76 ± 1.1 μΜ (Figure 7, Table 2). According to the ITC measurements, all the interactions of the La polypeptides are enthalpy-driven (Appendix A, Table 2), suggesting significant hydrogen bonding and Van der Waals interactions between the protein and RNA partners, and the determined K*_D_* values were the lowest for the larger La polypeptides, bearing the RRM1 domain, while the entire C-terminal sequence increased the RNA binding affinity over the shorter SBM segment (Table 2). ITC data indicate a single binding event for each interaction, and the fitting was made under this assumption.

## 3. Discussion

The La protein can bind different RNA motifs in the cytoplasm including the IRES domain of viruses, which exhibit tRNA mimicry [34]. Both the NMR and ITC results confirm the interaction and binding between the La protein and IRES domain IV of HCV, demonstrating the role of the hLa RRM2 domain. Additionally, the significant difference observed between the interaction studies of La C-terminal polypeptide with HCV-IRES and polyU_10_, manifested by the CSP analysis, indicates the specificity of the RRM2 domain for HCV-IRES and illuminates the RRM2 structural elements, which are important for this interaction. Collectively, our data suggest that the RRM2 domain of the human La protein exhibits the typical features of an RRM domain with the addition of an α-helix at its C-terminal sequence, and this folding also renders RNA binding properties in RRM2, although with different RNA specificity or recognition properties. Another similar RRM architecture is RRM1 of the polypyrimidine-tract binding protein 1 (PTBP1). Recent studies on PTBP1 show that the C-terminal α3 helix of this RRM domain acts as a sensor of the RNA secondary structure and is also crucial to promote Encephalomyocarditis virus IRES activity by PTB [35].

To unveil the role of the RRM2 domain in HCV-IRES, we compared the residues of RRM2 that are affected in all NMR titration experiments. The interaction site of RRM2 is the same according to the CSPs, and, when the RRM1 and/or C-terminus region is present, the RRM2 NHs involved in this process are more affected or disappear (Figure 5B). When RRM1 is present with the RRM2-SBM polypeptide, more NHs (from 27 residues) change their intensity or their chemical shifts in the RRM2-SBM region, indicating that the synergy of the two domains enhances HCV-IRES binding to the La protein. According to the chemical shift changes observed, RRM2 *β*2 and *β*3 strands possess a key role in the RRM2 capacity for HCV-IRES binding. The K*_D_* value for the interaction between RRM2-SBM and HCV-IRES is 28.2 ± 1.21 μΜ and drops to 10.3 ± 2.04 μΜ for the interaction between RRM1-RRM2-SBM and HCV-IRES. Interestingly, RRM2 does not show a binding preference for polyU_10_ RNA molecules. Instead, RRM1 alone or in the presence of LaM and/or RRM2 binds to poly(U)_10_ (unpublished data, [10]) utilizing, among others, the LaM *α*-helixes, whereas RRM1 binds to HCV-IRES via a part of all the *β-*strands. According to our experimental data, the role of RRM1 in the La’s RNA recognition and binding is twofold: (a) it binds RNA with the UUU-3′-OH tail of the RNA pol III transcripts, and (b) it participates in the La interaction with HCV-IRES, amplifying the binding properties of the RRM2 domain and the La’s C-terminal sequence.

Analysis of the kinetics along with the thermodynamics of the RRM2 interaction further support the specificity of the La C-terminal polypeptide toward HCV-IRES. In both RRM2-SBM and RRM2-Cter, residues found to participate in RNA binding and/or specificity events were located in the *β*2 and *β*3 strands and *α*1 and *α*3 helixes (Figure 3A and Figure 4A). When it comes to the titration of RRM1-RRM2-SBM with IRES, NMR studies suggest that both RRMs use the *β*-sheet surface to bind the RNA (Figure 5A), thus indicating the synergy of RRM1 and RRM2 for HCV-IRES, recognition and binding. The fact that most of the affected La polypeptide residues (42 out of 69) in RRM1-RRM2-SBM–HCV-IRES interaction are located in the RRM2-SBM region greatly supports the major contribution of RRM2 in the above process. Moreover, the present study reveals that in all the La-HCV-IRES titration experiments (RRM2-SBM–HCV-IRES, RRM2-Cter–HCV-IRES and RRM1-RRM2-SBM–HCV-IRES), the human La RRM2 α3 helix residues’ NHs are either perturbed or disappeared, underlying their role in RNA recognition and binding. Both the NMR conformational dynamics data and thermodynamics by ITC presented in this study indicate that the presence of the entire C-terminus region is of great importance for the interaction of the RRM2 polypeptide with HCV-IRES. Indeed, the K*_D_* values determined for the four La polypeptides show that (a) RRM1 increases the binding affinity for the IRES-HCV, and, (b) when the La polypeptides bear the entire C-terminal sequence, they exhibit a higher binding affinity compared to the La polypeptides included only with the SBM segment. These thermodynamic data are also in agreement with the NMR findings [7]. The data presented herein provide novel experimental evidence about the role of the RRM2 *α3* helix in La—HCV-IRES interaction and insights into the modulatory role of the La protein during translation regulation upon viral infection.

## 4. Materials and Methods

### 4.1. Sample Preparation

All the polypeptides of the La protein (hRRM1; residues 105–202, hRRM2; residues 225–334, hRRM2-SBM; residues 225–359, hRRM2-Cter; residues 225–408, hRRM1-RRM2-SBM; residues 105–359 and hRRM1-RRM2-Cter; residues 105–408) were cloned into pET.20b(+) vector and expressed in *E.coli* BL21(DE3) Rosetta 2 pLysS cells. The recombinant polypeptides were purified using standard IMAC affinity and size-exclusion chromatography protocols. Isotope enrichment for NMR spectroscopy was achieved in M9 minimal medium supplemented with 1 g/L ^15^N NH_4_Cl and ^13^C D-glucose, according to the experimental procedure described previously [14,23]. For the NMR and the ITC measurements, the buffer we used was 50 mM KPi, pH7. The solvent system of all the protein samples for the NMR measurements contained 90% H_2_O and 10% D_2_O.

### 4.2. Nuclear Magnetic Resonance (NMR) Spectroscopy

NMR measurements for all polypeptides (human RRM2-Cter and titration experiments) were performed at 298 K with a Bruker Advance III High-Definition, four-channel, 700 MHz spectrometer, equipped with a cryogenically cooled 5 mm ^1^H-^13^C/^15^N/D Z-gradient probe. The concentration of these polypeptides was 1mM. For RRM2-Cter, sequence specific assignments were obtained from the following heteronuclear experiments: 2D [^1^H–^15^N]-HSQC and TROSY version, 3D TROSY HNCA, 3D TROSY HN(CO)CA, 3D TROSY CBCA(CO)NH, 3D TROSY CBCANH, 3D TROSY HNCO, 3D TROSY HN(CA)CO, 3D HNHA and 3D 15N-NOESY.

All NMR data were processed with the TopSpin 3.2/3.5 software [36] and analyzed using Computer Aided Resonance Assignment (CARA) [37]. Dihedral angles with which we predicted the secondary structure of each construct were derived from TALOS+ [38].

### 4.3. Titration Experiments

To interpret the behavior of individual amino acids of each ^15^N-labeled La domains in the presence of unlabeled ligand, which is IRES sequence of hepatitis C virus (5′-AGACCGUGCACCAUGAGCACGAAUCCA-3), we calculated the changes of their chemical shifts in ^1^H–^15^N HSQC spectra during the NMR titration experiment. The concentration of each polypeptide was 0.1 mM. ^1^H-^15^N HSQC spectra were recorded at protein:RNA ratios of 1:0, 1:0.25, 1:0.5, 1:0.75, 1:1 and 1:1.5. No pH changes were detected before and after the RNA addition. Protein concentration has not changed because of the small amount of the concentrated HCV-IRES/polyU_10_ solution. The HCV-IRES stock concentration for the titration experiments was 5 mM, while the polyU10 stock concentration for the titration experiments was 2.5 mM. Chemical shift perturbation values were calculated using the following equation [39,40,41,42]:(1)$$\Delta \delta_{AV}=\sqrt{\Delta \delta_{AV}{\left({\;}^{1}HN\right)}^{2}+{\left[\frac{1}{5}\times \Delta \delta_{AV}\;\left({\;}^{15}N\right)\right]}^{2}}$$

The threshold value in all titration experiments was defined by using the standard deviation (*σ*) of the CSPs and then multiplying by 2 [39].

All NMR data were processed with the TopSpin 3.2/3.5 software [36] and analyzed using Computer Aid Resonance Assignment (CARA) [37]. Chemical shifts for the analysis of titration experiments come from our resonance assignment analysis of LaM-RRM1 and RRM2-Cter. For RRM1-RRM2-SBM–HCV-IRES, chemical shifts for the interRRM linker were obtained from the deposit resonance assignment of LaM-RRM1-RRM2 (BMRB code: 17878) [6].

### 4.4. Protein Dynamics

The backbone dynamics of all polypeptides were studied through the analysis of ^15^N *R_1_* and *R*_2_ relaxation rates and heteronuclear {^1^HN}-^15^N NOEs. A series of 2D ^1^H-^15^N HSQC spectra in a pseudo-3D mode were recorded for *R*_1_ relaxation rate, with relaxation delays of 20 ms, 60 ms, 100 ms, 200 ms, 400 ms, 600 ms, 800 ms and 1200 ms for RRM2-Cter. A series of 2D ^1^H-^15^N HSQC spectra were recorded for *R*_2_ relaxation rate as well. *R*_2_ relaxation delays for RRM2-Cter were 15.68 ms, 31.36 ms, 62.72 ms, 94.08 ms, 125.44 ms, 156.80 ms, 188.16 ms and 219.16 ms.

The chemical shifts of RRM2-Cter were obtained through our NMR analysis (BMRB code: 51713) All data of backbone dynamics for RRM2-Cter were analyzed with Bruker’s Dynamic Center software.

### 4.5. Isothermal Titration Calorimetry (ITC)

Isothermal Titration Calorimetry (ITC) measurements were performed at 25 °C on MicroCal PEAQ-ITC calorimeter (Malvern Panalytical, Malvern, UK). For all the ITC experiments, protein and RNA solutions were prepared in 50 mM KPi at pH7. In each titration, 13 injections of 3 μL each of the HCV-IRES solution (500 μM) were made into the sample cell containing the protein solution (50 μM). A spacing of 150 s between each injection was applied to enable the system to reach equilibrium. All the experiments were performed in duplicate. The ITC data were analyzed with Microcal-PEAC-ITC Analysis software. Binding parameters such as the K*_D_* (dissociation constant), the Δ*H* (enthalpy change) and the Δ*S* (entropy change) were determined by fitting the experimental binding isotherms with the appropriate models. The total heat content *Q* of the solution contained in *V*_0_ (determined relative to zero for the unliganded species) at fractional saturation *Θ* (fraction of sites occupied by the ligand) is
(2)$$Q=n\Theta \;M_{t}\Delta {HV}_{0}$$
where
(3)$$\Theta =\frac{1}{2}\left[1+\frac{X_{t}}{{nM}_{t}}+\frac{1}{{nKM}_{t}}-\sqrt{{\left(1+\frac{X_{t}}{{nM}_{t}}+\frac{1}{{nKM}_{t}}\right)}^{2}-\frac{{4X}_{t}}{{nM}_{t}}}\right]$$

Substituting the *Θ* parameter into the equation above, gives
(4)$$Q=\frac{{nM}_{t}\Delta {HV}_{0}}{2}\left[1+\frac{X_{t}}{{nM}_{t}}+\frac{1}{{nKM}_{t}}-\sqrt{{\left(1+\frac{X_{t}}{{nM}_{t}}+\frac{1}{{nKM}_{t}}\right)}^{2}-\frac{{4X}_{t}}{{nM}_{t}}}\right]$$
where *n* is the number of sites, *M* is the free concentration of macromolecule in the active volume, *K i*s the binding constant, and Δ*H* is the molar heat of ligand binding. The heat released or absorbed during the *i*th injection is given by
(5)$$\Delta Q\left(i\right)=Q\left(i\right)+\frac{{dV}_{i}}{V_{0}}\left[\frac{Q\left(i\right)+Q\left(i-1\right)}{2}\right]-Q\left(i-1\right)$$
where *V_i_* is the volume of the *i*th injection, *V*_0_ is the cell volume, and *Q*(*i*) is the heat following the *i*th injection.

## Figures and Tables

**Figure 1 ijms-24-02572-f001:**
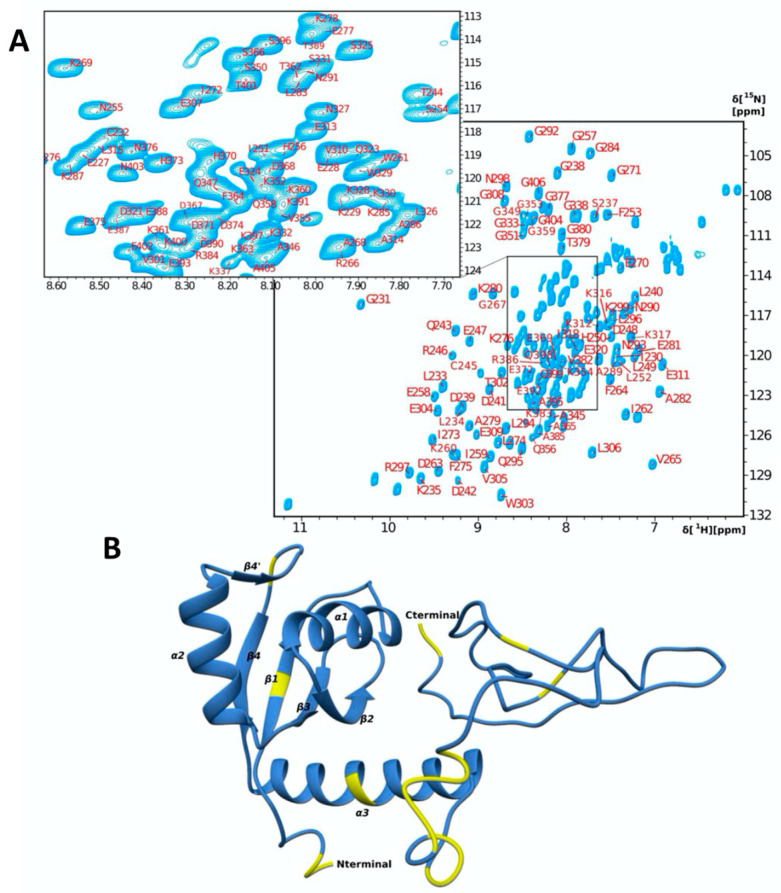
(**A**) ^1^H-^15^N HSQC spectrum and assignment of *human* La RRM2-Cter; (**B**) chemical shift mapping of the unassigned residues’ NHs (yellow) of the CS Rosetta model of RRM2-Cter.

**Figure 2 ijms-24-02572-f002:**
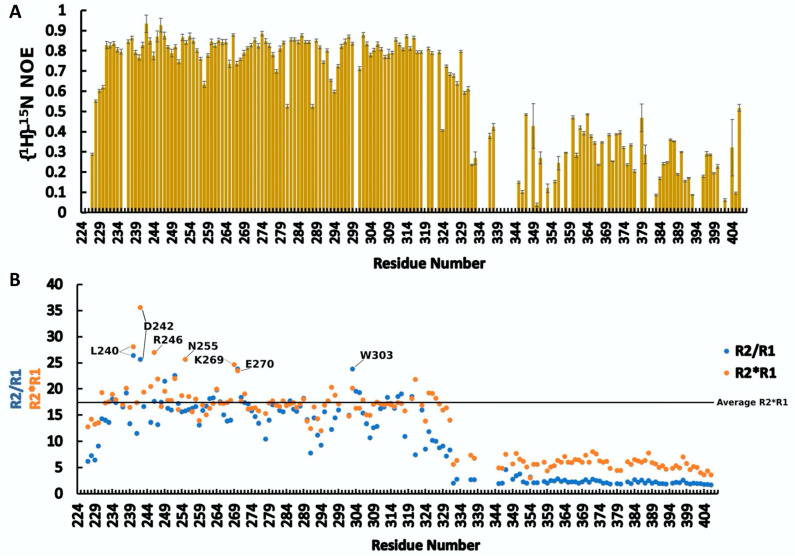
(**A**) *heteroNOE* values of RRM2-Cter (La 224–408); (**B**) *R*_2_*/R*_1_ and *R*_1_
*× R*_2_ values of RRM2-Cter (La 224–408). Residues with extremely high values of *R*_1_
*× R*_2_ and/or *R*_2_*/R*_1_ are noted.

**Figure 3 ijms-24-02572-f003:**
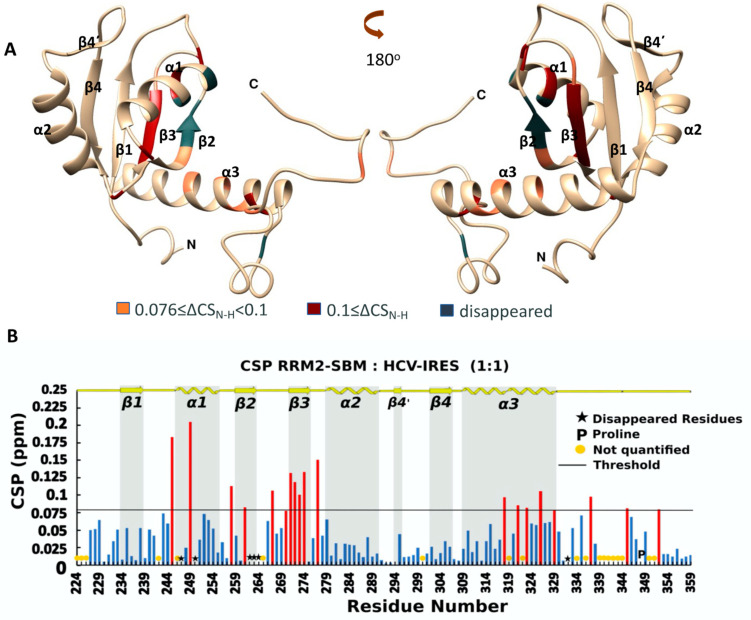
(**A**) Ribbon representation of the affected amino acids mapped into the hRRM2-SBM structure; (**B**) chemical shift perturbation of ^15^N-hRRM2-SBM upon the addition of the unlabeled ligand in threshold value 0.076. Red bars indicate CSPs above threshold, while blue bars illustrate CSPs beyond threshold line.

**Figure 4 ijms-24-02572-f004:**
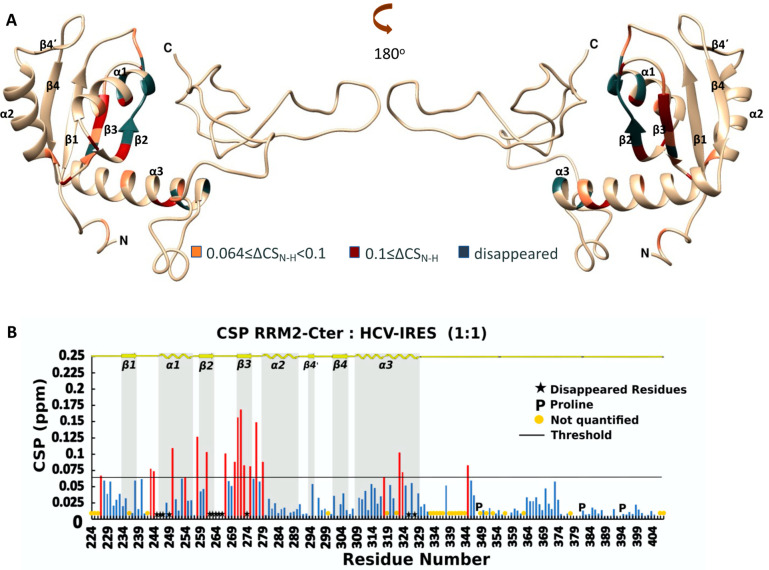
(**A**) Ribbon representation of the affected amino acids mapped into the hRRM2-Cter structure; (**B**) chemical shift perturbation of ^15^N-hRRM2-Cter upon the addition of the unlabeled ligand in threshold value 0.064. Red bars indicate CSPs above threshold, while blue bars illustrate CSPs beyond threshold line.

**Figure 5 ijms-24-02572-f005:**
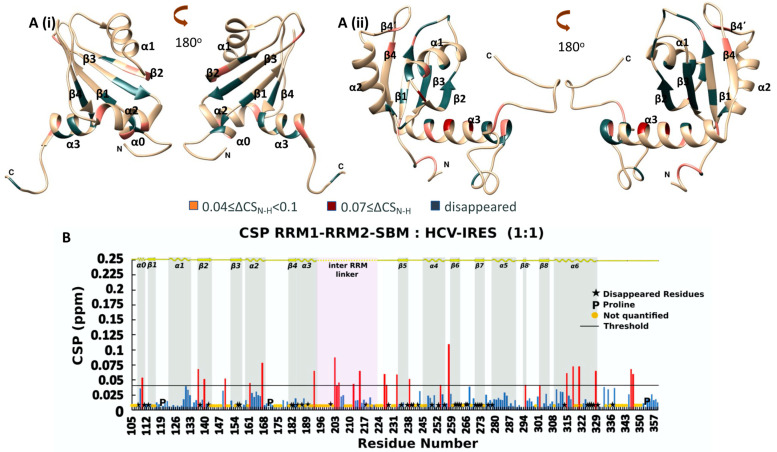
(**A**) Ribbon representation of the affected amino acids mapped into hRRM1 (*i*) (PDB ID: 1S79) and the 3D structure of hRRM2-Cter (*ii*) predicted by CS Rosetta; (**B**) chemical shift perturbation of ^15^N-hRRM1-RRM2-SBM upon the addition of the unlabeled ligand in threshold value 0.04. Red bars indicate CSPs above threshold, while blue bars illustrate CSPs beyond threshold line.

**Figure 6 ijms-24-02572-f006:**
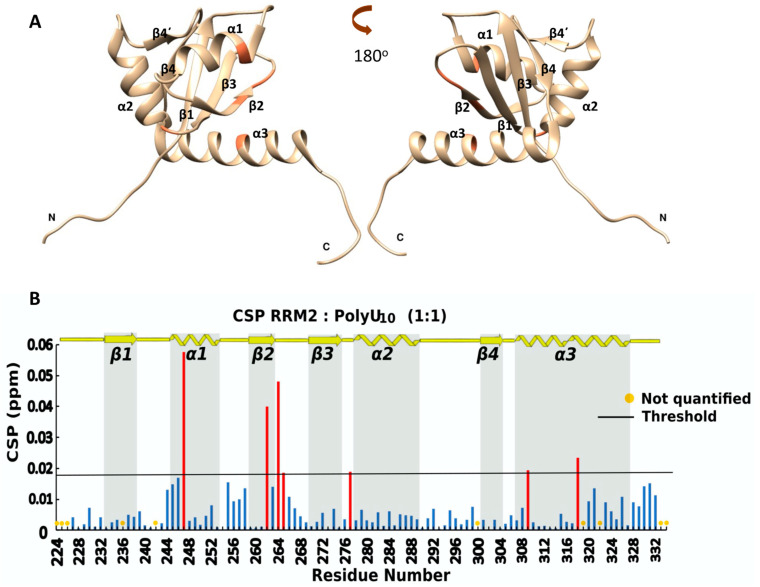
(**A**) Ribbon representation of the affected amino acids mapped (in orange) into the hRRM2 structure (PDB ID:1OWX); (**B**) chemical shift perturbation of ^15^N-hRRM2 upon the addition of the unlabeled ligand in threshold value 0.018. Red bars indicate CSPs above threshold, while blue bars illustrate CSPs beyond threshold line.

**Figure 7 ijms-24-02572-f007:**
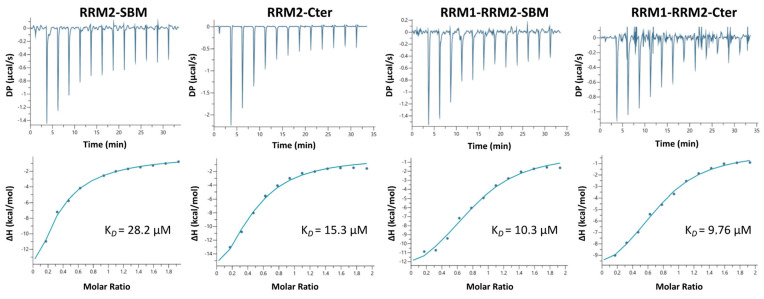
ITC analysis of the interactions between the human La polypeptides (RRM2-SBM, RRM2-Cter, RRM1-RRM2-SBM and RRM1-RRM2-Cter) and IRES-HCV domain IV. The upper graph in each plot displays the raw data, and the lower graph displays the intergraded heat per injection (circles) and the best fit function (solid line). K*_D_* values are presented for each interaction.

**Table 1 ijms-24-02572-t001:** Average values of *R*_1_, *R*_2_ and *heteroNOE* parameters of different parts of RRM2-Cter (La 224–408).

	*R* _1_	*R* _2_	*heteroNOE*
RRM2-Cter (aa 224–408)	1.29	11.6	0.62
RRM2 (aa 224–334)	1.13	16.11	0.77
C terminal (aa 335–408)	1.59	3.61	0.34

**Table 2 ijms-24-02572-t002:** Thermodynamic parameters of the interactions between the human La polypeptides and IRES-HCV domain IV.

	K*_D_*(μM)	Δ*H* (kcal/mol)	−*Τ*Δ*S* (kcal/mol)	Δ*G* (kcal/mol)
RRM2-SBM	28.2 ± 1.21	−80.0 ± 1.89	73.79	−6.21
RRM2-Cter	15.3 ± 1.49	−27.2 ± 0.99	20.63	−6.57
RRM1-RRM2-SBM	10.3 ± 2.04	−15.1 ± 1.13	8.28	−6.80
RRM1-RRM2-Cter	9.76 ± 1.1	−12.04 ± 0.36	5.2	−6.84

## Data Availability

The data that support these findings are available upon request from the corresponding author.

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
