# Peer review of "NMR Analysis Suggests Synergy between the RRM2 and the Carboxy-Terminal Segment of Human La Protein in the Recognition and Interaction with HCV IRES"

_ijms, 2023, doi:10.3390/ijms24032572_

Round 1

Reviewer 1 Report

I have gone through the manuscript entitled "NMR analysis suggests synergy between the RRM2 and carboxy-terminus segment of human La protein in the recognition and interaction with HCV IRES" for publication in this journal. I found that all the sections of the manuscript are presented well and the work done by the authors is new and has scientific significance and innovation.

Therefore, I recommend the publication of this manuscript in the International Journal of Molecular Sciences after minor corrections as given below:

1.      Please check the entire manuscript for grammatical errors.

2.      The introduction must be improved and recent work should be discussed.

3.      The aim of the paper should be a little more highlighted in the last paragraph of the introduction.

4.      Go through the following recent articles and cite them accordingly- DOI: 10.3390/ph15030285; 10.1039/D1ME00147G; 10.1007/s11224-022-01996-y; 10.1016/j.bioorg.2021.105572; 0.1016/j.molliq.2021.118121

5.      Molecular docking, MD Simulation, or other ab initio quantum chemistry methods could be used (can go through the above mentioned articles for reference).

6.      Conclusion should be rewritten.

7.      Please increase the resolution of the figures. Labels are not visible.

Author Response

Response to Reviewer’s 1 Comments

Point 1: Please check the entire manuscript for grammatical errors.

Response 1: Manuscript is thoroughly revised for grammatical errors

Point 2: The introduction must be improved and recent work should be discussed.

Response 2: Introduction is now revised and recent work is now cited and discussed

Point 3: The aim of the paper should be a little more highlighted in the last paragraph of the introduction.

Response 3: The aim of the paper has been revised and highlighted according to the reviewer’s suggestion

Point 4: Go through the following recent articles and cite them accordingly- DOI: 10.3390/ph15030285; 10.1039/D1ME00147G; 10.1007/s11224-022-01996-y; 10.1016/j.bioorg.2021.105572; 0.1016/j.molliq.2021.118121

Response 4: Unfortunately, we do not see any scientific relationship of these articles, neither with the experimental part nor with the scientific section of our work. Theoretical calculations are irrelevant to our work since we do not need to perform MD simulation or ab initio calculations to build a protein-RNA complex. First, ab initio calculations for such big biomolecular complex. Second MD simulation to protein RNA complex may provide some kind of information that are not relevant with our study to prove the selectivity of the La protein and not the stability of any complex formed. Additionally, and most importantly, this work does not provide, because it is not its aim, any information on the RNA binding site. Finally, we do not find that theoretical studies suggested, will add any value to our wealth of experimental evidence obtained by NMR & ITC.

Point 5: Molecular docking, MD Simulation, or other ab initio quantum chemistry methods could be used (can go through the above mentioned articles for reference).

Response 5: Same as above. Please our response on Comment #4.

Point 6: Conclusion should be rewritten.

Response 6: Conclusions are revised accordingly

Point 7: Please increase the resolution of the figures. Labels are not visible.

Response 7: Figures are revised accordingly

Reviewer 2 Report

La is one of the popular RNA-binding proteins (RBPs) that bind and regulate RNAs1. However, the recognition and interaction mechanism between La and RNA is not clear. Argyriou et al. present their study of human La protein interacting with RNA in solution. Through NMR titration and ITC study of different La protein constructs, including RRM2-SBM, RRM2-Cter, RRM1-RRM2-SBM and RRM1-RRM2-Cter, they proved that the C-terminal domain of La protein can recognize and bind with HCV IRES specifically. Using NMR to investigate the precis binding sites between protein and RNA is a common method. The novelty in this study is they identified RNA binding sites at individual residue level, which cannot be achieved by other binding experiments, such as gel shift assays and ITC. This will be highly interested by researchers who are working on drug design or functional study. Overall, the manuscript is clearly written. The methods and results are sound.

Comments/suggestions/questions:

1.     In 2.1, the statement "Although the resonance assignment for the La RRM2 is reported in the literature 77 (BMRB accession no 5235), NMR characterization of a larger polypeptide, that spans the 78 RRM2 and the C-terminal (Cter) segment, is not available." needs to be modified. The NMR spectrum of RRM2-Cter has been reported in reference2 in the main text, however the assignment of RRM2-Cter is missing.

2.     In 2.2, could the authors explain more about how the average values of R1, R2 and hnNOE demonstrate the rigid of the RRM2 domain? The traditional way to analyze R1, R2, hnNOE data is plotting R2/R1 and R1*R2 figures to identify whether residues undergoing chemical exchange3. Thus, it can provide information about the rigid regions as well as flexible regions.

3.     In 2.2, some residues in Cter show large hnNOE values, similar to the values shown in RRM2. Do these residues have interaction with RRM2? Could the authors provide some explanation?

4.     In Figure S3, some residues located round residue 240 and residue 268 show large R2 values. Are these residues involved in the binding pocket? Please plot R2/R1 figure and explain the residues with large R2/R1 values.

5.     In 2.4, only 7 residues show large CSP values after adding polyU10. How about the intensity before and after adding polyU10? As we know that some binding reactions may not change the chemical shift but change the intensity. It's highly suggested to do intensity analysis along with CSP analysis.

6.     In 2.4, "Figure 10" should be "Figure S10".

7.     In 4.3, "Protein concentration has not changed because of the small amount of the concentrated HCV-IRES/polyU10 solution." Please provide the HCV-IRES/polyU10 stock concentrations for these titration experiments.

8.     In 4.5, please provide the model (equation?) used to fit the ITC experimental data.

References:

1.         M. Corley, M. C. Burns and G. W. Yeo, Mol Cell, 2020, 78, 9-29.

2.         A. Jacks, J. Babon, G. Kelly, I. Manolaridis, P. D. Cary, S. Curry and M. R. Conte, Structure, 2003, 11, 833-843.

3.         J. M. Kneller, M. Lu and C. Bracken, J Am Chem Soc, 2002, 124, 1852-1853.
